# Questionnaire-Based Study Evaluating the Hand Hygiene Practices and the Impact of Disinfection in the COVID-19 Pandemic on Hand Skin Conditions in Poland

**DOI:** 10.3390/jcm12010195

**Published:** 2022-12-27

**Authors:** Agnieszka Polecka, Natalia Owsianko, Andrzej Awchimkow, Anna Baran, Justyna Magdalena Hermanowicz, Iwona Flisiak

**Affiliations:** 1Department of Dermatology and Venerology, Medical University of Bialystok, Zurawia 14 St., 15-540 Bialystok, Poland; 2Department of Pharmacodynamics, Medical University of Bialystok, Mickiewicza 2C St., 15-089 Bialystok, Poland

**Keywords:** hand eczema, COVID-19, SARS-CoV-2, coronavirus, pandemic, hand hygiene, disinfection, skin of the hands

## Abstract

During the COVID-19 pandemic, disinfection became an integral part of everybody’s life in order to avoid spreading the coronavirus. In 2021, an original anonymous online survey was carried out. The questions concerned the usage of disinfectants. The study population included 56 subjects diagnosed by a physician with hand eczema (HE-derm group) and 114 subjects with no hand skin disease diagnosed by a specialist (non-derm). The HE and non-HE groups were distinguished. Nearly 80% of the HE group, and 10% of the non-HE group, experienced worsening of hand skin lesions caused by increased skin disinfection. HE-group respondents more often declared the occurrence of new hand skin symptoms, over 80% of the subjects of this group had more than 1 new symptom (compared to nearly 40% of the non-HE group). Exacerbations of the skin disease were more frequently observed by the HE group during the pandemic. There was a statistically significant decrease of the quality of life in the HE group compared to the non-HE group during the pandemic. The COVID-19 pandemic caused an increase in the prevalence of hand skin symptoms and deterioration of the skin condition. Education on appropriate disinfection techniques and skincare, as well as early dermatological intervention, might allow us to limit the development of hand skin diseases.

## 1. Introduction

Since 2019, healthcare activities have focused on fighting the COVID-19 pandemic caused by the SARS-CoV-2 virus, which has become one of the most widespread viruses of the 21st century in the world. Infection with the new coronavirus mainly affects the respiratory and digestive systems, but complications of the disease can involve various organs, including the skin. The pandemic has affected the life, health, emotional and psychosocial state of the entire population. It has forced changes in many habits and limited contact with relatives. In order to prevent transmission of the virus, the World Health Organization (WHO) currently recommends different preventive measures, including maintaining social distance, obligatory use of personal protective equipment (PPE), frequent hand washing using disinfectants or measuring temperature in public places [1]. With regard to new guidelines, there has also been a significant increase in the use of alcohol-based skin disinfectants and a more frequent use of cleansers, because shortly after the outbreak of the pandemic, isopropylene alcohol, with a minimum concentration of 70%, was proven to be virucidal against SARS-CoV-2 [2]. Such frequent application of alcohol and surfactants may contribute to the violation of the physiological hydro-lipid barrier of the epidermis and disturb the microbiome of the skin, consequently causing the appearance of new or exacerbation of already-diagnosed skin diseases. Considering the recommendations of the WHO or CDC (Centers for Disease Control and Prevention) suggesting frequent hand washing using soap and water for 20 s or, alternatively, if soap and water are unavailable, hand sanitizer containing at least 60% alcohol in order to reduce the risk of spreading infection with the new virus, it should be kept in mind that these may have negative consequences in the form of deterioration of the natural flora and skin barrier what may increase the incidence of a large group of patients in dermatological practice with not only new-onset hand eczema (HE) but also exacerbation of other hand skin dermatoses [3,4].

The term “eczema” refers to a specific kind of inflammatory skin disorder that affects both the epidermis and the dermis [5]. Skin inflammation that only affects the hands and/or wrists is known as hand eczema [6]. Depending on the stage of the disease, it presents a certain pattern of histological and clinical characteristics [7]. The most common primary symptoms of HE are erythema, oedema, papules, and vesicles. Hyperkeratotic lesions, scaling, and fissures are examples of secondary chronic lesions. The most prevalent symptom of all eczema kinds is pruritus [5].

In the general population, hand eczema affects 14.5% of persons over their lifetime, with a pooled incidence rate of 7.3 cases/1000 person-years [8]. The occurrence of HE appears to have been increasing in recent years. In addition, female sex is related to an earlier age of HE onset [8]. The most significant risk factor for HE has been established as being atopic dermatitis (AD), which is associated with a three- to four-fold increase in the prevalence of HE compared to the general population [9,10]. Other risk factors for developing HE are low age at onset of HE, being contact allergic (e.g., nickel, cobalt, fragrance mixes I and II, and formaldehyde), being exposed to wet work, and cold or dry weather conditions [5,11]. Due to healthcare workers’ (HCWs) exposure to significant amount of wet work, they are more prone to develop HE. The prevalence of HE among healthcare professionals before the COVID-19 pandemic varied from 13% to 65% depending on regions or countries (data from Germany and USA, respectively) [12]. Educational intervention for HCWs resulted in improving knowledge and inducted behavioral change, such as the reduction of frequency of handwashing [13]. Additionally, disinfecting agents used in operating rooms often have a complex formulation and can lead to contact dermatitis [14]. During the pandemic, HCWs have led to increased hygiene behaviors, both in private life and at work. Hand washing 8–10 times per day is sufficient to considerably increase the risk of hand eczema compared to an individual who washes their hands less frequently [6].

There are various HE treatment methods. Topical anti-inflammatory agents (topical corticosteroids, calcineurin inhibitors) and emollients, as well as systematic treatment (oral corticosteroids, alitretinoin, acitretin, methotrexate, cyclosporine, azathioprine, and biological therapy) are recommended to control HE symptoms [5]. The increased frequency of emollient use at work and at home has led to improved skin condition reported by individuals with HE [15]. AD is a chronic inflammatory skin disease. It is a highly pruritic skin condition with erythematous lesions [16]. Atopic dermatitis is associated with other allergic conditions and is often one of the first signs of “atopic march” [17]. Asthma or allergic rhinitis are typical among AD patients. There are several triggering factors, including stress, some allergens, and infections [18]. The prevalence of AD has been increasing in recent years, it affects up to 30% of children and 10% of the adult population [19].

Psoriasis is a chronic skin disease mediated by the immune system, with a prevalence of 2% worldwide [20]. This condition affects the skin, joints, and nail, and significantly decreases the quality of patients’ lives [21]. Approximately 50% of individuals with cutaneous manifestations develop nail changes [20]. Nail psoriasis is challenging to treat and worsens the patients’ outcomes.

To the best of our knowledge, no studies have been locally conducted and published regarding the disinfection impact on the prevalence of hand eczema among the Polish population during COVID-19. We hypothesized that hand skin deterioration might be increased due to more frequent washing and disinfection, not only in the general population but especially in persons with previous hand skin dermatoses. The aim of the study was to evaluate the hygiene and hand care attitudes in the pandemic era in comparison to the time before its outbreak. Furthermore, we investigated whether the recommendations for hand sanitation and disinfection exacerbate hand eczema or favor the development of new hand skin problems in the general population as well as in already-diseased subjects; additionally, we intended to identify the factors associated with the exacerbation of hand skin eczema and point to helpful hygiene tips in order to improve the hand skin barrier.

## 2. Materials and Methods

### 2.1. Ethical Approval and Informed Consent

This study was approved by the University Bioethics Committee (Ref.-No. APK.002.469.2020) and was in accordance with the Helsinki Declaration. All participants were informed about the conducted research and data collection. All respondents provided informed consent. Participation in the study was voluntary. Participants could withdraw their consent at any time during the survey.

### 2.2. Study Design and Population

This was a descriptive, observational, original, online anonymous questionnaire survey evaluating the impact of disinfection on the condition of the skin of the hands and quality of life, especially in people with dermatoses, before and during the COVID-19 pandemic.

### 2.3. Study Protocol

This study was conducted in the form of a personalized, original online questionnaire created by the authors of the study at the Department of Dermatology and Venereology, Medical University of Bialystok, Poland, from January to March 2021. The survey was addressed to all people over 16 years old, including those with hand skin dermatoses. The group of respondents consisted of 170 subjects.

The online questionnaire, designed in Polish language, using Google Forms, was electronically distributed on 11 January 2021 via social media and applications (Facebook, WhatsApp) to people working or studying at medical faculties, as well as groups of people suffering from skin diseases. The time taken to collect and analyze the data was 2 months.

The original questionnaire for assessing the impact of disinfection on the condition of the skin of the hands during the pandemic was divided into 4 parts. The first part contained basic information on gender, age, education, the field of studies or profession, and demographic data. The second part concerned dermatological history, including previous history of atopy or hand skin diseases and current state of the skin of the hands. Questions about the diagnosis of skin lesions, such as biopsy or epidermal patch tests and their results with an indication of allergenic factors, were also included. The second part also included questions about the treatment of lesions occurring on the skin of the hands, as well as the accompanying symptoms during exacerbations of the disease.

The third part of the original survey concerned the identification of new symptoms or the severity of hand skin diseases as comparative data before and during the COVID-19 pandemic, as well as the impact of disinfection on the quality of life of the individuals, especially with hand skin dermatoses. The questions analyzed daily hand hygiene, such as frequency and volume of disinfectants usage, the accuracy of hand drying before each application of the disinfectant, and the appearance of specific symptoms after the application of disinfectants, such as pain, burning, dryness, itchiness, swelling, redness, or increased warmth. The third part also collected data on possible changes in the method and duration of treatment of skin lesions during the COVID-19 pandemic, changes in the frequency of visits to dermatologists, as well as the duration of remission periods during the pandemic, compared to those before recommendations of frequent skin disinfection. Data were also collected on skin moisturizing habits before and during the pandemic.

In order to determine the nuisance of disinfection and the quality of life in the era of the SARS-CoV-2 pandemic, a five-step visual analog scale was used, where 1 meant that disinfection had no negative effect on the condition of the skin of the hands and quality of life, and 5 indicated a significant impact of disinfection on the condition of the skin of the hands and a lowering of quality of life. The questions prepared by the authors were reviewed by two expert dermatologists for consistency and appropriateness. The questions included in the survey are presented in Appendix A in Appendix A.

### 2.4. Statistical Analysis

Data were collected in Excel, and statistical analyses were performed with the GraphPad Prism 9.20 software (GraphPad Prism 9.20 Software, USA). Descriptive analysis was performed by calculating the frequencies and percentages of variables. The relationships between two variables were analyzed using the chi-square independence test. *p* values < 0.05 were considered statistically significant.

## 3. Results

### 3.1. Study Group

A total of 170 adult participants took part in an anonymous survey, 142 were women (83.53%) and 28 were men (16.47%). Among the respondents, 56 participants reported hand eczema diagnosed by physician (32.94%), including 10 participants (5.88%) who declared no dermatological symptoms before the outbreak of the pandemic (Figure 1).

The 56 participants who were diagnosed by physician with hand eczema were classified as the HE-derm group (Figure 2).

The second study group, classified as non-derm, included 114 respondents (67.06%) who have never been diagnosed by physician with hand skin diseases. The non-derm group was divided in two subgroups: non-HE (n = 68/114, 59.65%), in which no respondent declared the appearance of any symptoms of hand eczema, and HE n/derm (n = 46/114, 40.35%), in which symptoms of hand eczema appeared. Analyzing both groups (non-derm and HE-derm) allowed for the HE-study group (102/170, 60.00%) to be distinguished. These participants were distinguished according to the reported symptoms as individuals with hand eczema. We chose to place these participants in the HE-study group based on the definition of HE regardless of the fact that they had not seen the physician but the symptoms mentioned by the respondents lasted longer than expected and did not disappear spontaneously [5]. The results are, therefore, more precise and responses from the study participants are more accurately evaluated.

The demographic data of the study group are demonstrated in Table 1.

The most commonly reported dermatosis in the HE-derm group was AD (n = 33, 58.93%) (Table 2).

### 3.2. Study Outcomes

Out of the participants from the HE-study group currently experiencing hand skin lesions, 56/102 (54.90%) reported that, before the COVID-19 pandemic, they had hand skin lesions, while 46/102 (45.10%) persons had no hand skin problems before the outbreak of the pandemic.

The subject-reported factors that most frequently caused skin lesion exacerbation were detergents in 55.88% (57/102), latex in 25.49% (26/102), and metals in 18.63% (19/102).

During the pandemic, 42/46 (91.30%) participants with dermatological history experienced worsening of hand skin lesions caused by increased skin disinfection. Over half of the HE-derm group (24/46, 52.17%) reported new symptoms that they noticed after disinfection, and a significant proportion of this group (69.57%, n = 32) also stated they had more frequent exacerbations of skin lesions associated with increased hand skin hygiene. For comparison, 8 out of 10 participants (80%) from the HE-derm group, who first observed skin lesions during the pandemic, indicated that disinfection significantly worsened the course of the disease and caused more frequent exacerbations. The most frequent new symptom in 50% of this group (5/10) was unpleasant dryness of the hands.

In the HE-study group of 102 subjects, 79.41% (n = 81) reported that they had new symptoms and 60.78% (n = 62) noted exacerbations of skin lesions associated with increased skin hygiene.

Participants from the HE-study group complained of a greater number of new symptoms (most often dryness (62/102; 60.78%), roughness (53/102; 51.96%), and redness (40/102; 39.22%)), and noted symptoms that were not reported by subjects from the non-derm group: vesicles filled with serum fluid (7/102, 6.86%) and pustules (5/102, 4.90%). Respondents were allowed to select more than one symptom from those listed by the authors, so the results may exceed 100% of the total number respondents from both study groups (Table 3).

#### 3.2.1. The Occurrence of New Symptoms

The prevalence of the occurrence of new symptoms was significantly higher in the HE-study group compared to non-HE, HE-n/derm to non-HE, and HE-derm to non-HE during the pandemic (*p* < 0.001, *p* < 0.05, *p* < 0.05, respectively). There was no significant correlation between HE-n/derm and HE-derm with regard to the occurence of new symptoms (Figure 3).

83/102 (81.37%) respondents from the HE-study group and 27/68 (39.71%) out of non-HE group immediately after the application of an alcohol-based disinfectant experienced more than one symptom, most often it was dryness (HE-study group: 87/102 (85.29%); non-HE group: 37/68 (54.41%)), roughness (HE-study group: 59/102 (57.84%); non-HE group: 22/68 (32.35%)), and redness (HE-study group: 53/102 (51.96%); non-HE group: 15/68 (22.06%)) (Figure 4).

In addition to the aforementioned symptoms, respondents have also reported experiencing pruritus, increased skin warmth, and edema. Their prevalence is shown in Figure 5, Figure 6 and Figure 7.

The pruritis was observed significantly more frequent in the HE-study group compared to the non-HE study group, HE-derm to HE-n/derm, and HE-derm to non-HE (*p* < 0.05, *p* < 0.05, *p* < 0.05, respectively). There was no significant difference between non-HE and HE-n/derm with regard to the frequency of pruritis in these groups (Figure 5).

We have noticed that increased skin warmth was significantly more frequent in the HE-study group compared to the non-HE study group, HE-derm to HE-n/derm, and HE-derm to non-HE (*p* < 0.05, *p* < 0.05, *p* < 0.05, respectively). There was no significant correlation between non-HE and HE n/derm with regard to increased skin warmth in these study groups (Figure 6).

Edema was observed only in the HE-study group, in which 11 participants experienced this symptom. Most of the subjects (10/11, 90.91%) were members of HE-derm group, and 1 respondent was from the HE n/derm group. No participant mentioned the appearance of edema in the non-HE study group (Figure 7).

Additionally, further comparison of the following symptoms between study groups is presented in Appendix A of the Appendix A.

#### 3.2.2. Negative Effects and Lack of Negative Effects

Only 8.82% (n = 9/102) of the HE-study group and 38.24% (n = 26/68) of the non-HE group did not experience any negative effects on the condition of the skin during the application of the antiseptic (Figure 8).

86.27% of the subjects from the HE-study group (n = 88/102) reported pain and burning sensations after using disinfectants, while in the non-HE group, it was reported by 42.65% (n = 29/68) (Figure 9).

The frequency of pain and burning sensations was significantly higher in the HE-study group compared to the non-HE group, HE-n/derm to non-HE, and HE-derm to non-HE (*p* < 0.001, *p* < 0.0001, *p* < 0.0001, respectively). There was no significant difference etween HE-n/derm and HE-derm with regard to the frequency of pain and burning sensations in these study groups (Figure 10).

#### 3.2.3. The Exacerbations of Skin Lesions

62/102 participants (60.78%) from the HE-study group more often observed exacerbations in their underlying hand skin disease during the pandemic, and 10/68 (27.19%) of the non-HE group reported that disinfection-related symptoms caused more frequent and longer-lasting skin lesions (*p* < 0.05). (Figure 11).

During the pandemic, exacerbations of skin lesions were significantly more frequent in the HE-study group compared to non-HE, HE-derm to HE-n/derm, HE-n/derm to non-HE, and HE-derm to non-HE (*p* < 0.001, *p* < 0.05, *p* < 0.001, *p* < 0.001, respectively) (Figure 12).

#### 3.2.4. Medical Appointments

Additionally, 16 participants (15.69%) from the HE-study group had to seek medical advice more often during the pandemic, and there was one such person from the non-HE group (1.47%). (Figure 13).

Medical appointments were significantly more frequent in the HE-study group compared to non-HE, HE-derm to HE-n/derm, and HE-derm to non-HE (*p* < 0.01, *p* < 0.001, *p* < 0.001, respectively). There was no significant difference between non-HE and HE-n/derm in the frequency of medical appointments during the pandemic (Figure 14).

#### 3.2.5. New Treatment

During the pandemic, none of the participants from the non-HE group, who observed disinfection-related skin symptoms, reported usage of any treatment, while 14/102 participants (13.73%) from the HE-study group received more advanced therapy (e.g., changed from topical to systemic treatment, increased the treatment dosage, or new drug added). (Figure 15).

Treatment intensification was used significantly more often in the HE-study group compared to non-HE, HE-derm to HE-n/derm, and HE-derm to non-HE during the COVID-19 pandemic (*p* < 0.01, *p* < 0.001, *p* < 0.001, respectively). No difference between non-HE and HE-n/derm with regard to the changes in treatment was observed (Figure 16).

Additionally, a comparison of the treatment prolongation during the pandemic in the HE, HE-derm, and HE-n/derm study groups is presented in Appendix A of the Appendix A.

Appendix A of the Appendix A presents the duration of remission period in the HE-derm group.

#### 3.2.6. The Frequency of Hand Skin Disinfection

Over 15 of the non-HE group (20.59%, n = 14/68) disinfected their hands about 1–2 times a day, 66.18% (n = 45/68) several times a day, and 13.23% (n = 9/68) several times an hour. Comparing the HE-study group, 27.45% (n = 28/102) disinfected the skin about 1–2 times a day, 54.90% (n = 56/102) several times a day, and 17.65% (n = 18/102) disinfected several times an hour (Figure 17).

The vast majority of the HE-study (82.35%, 84/102) and non-HE (72.06%, 49/68) groups applied the appropriate volume, recommended by the manufacturer, of the disinfecting agents for the entire surface of the skin of the hands, and the remaining respondents used more of the disinfectant than recommended by the manufacturer. None of the study participants indicated using too little of the product (Figure 18).

#### 3.2.7. The Moisturizing

The questionnaire included questions assessing the frequency of moisturizing the skin of the hands before and during the SARS-CoV-2 pandemic. Before the pandemic, 94.12% (96/102) of HE-study group and 79.41% (54/68) of the non-HE group moisturized the skin of their hands. In the HE-study group, 29.41% (30/102) of participants used emollients several times a week, 36.27% (37/102) 1–2 times a day, and 34.31% (35/102) more than 3 times a day. In the non-HE group, these results were 45.59% (31/68), 26.47 (18/68), and 27.94% (19/68), respectively.

During the pandemic, 99.10% (101/102) of the HE-study group and 83.82% (57/68) of non-HE moisturized the skin of their hands. In the HE-study group, 18.81% (19/101) moistened the skin several times a week, 31.68% (32/101) 1–2 times a day, and 49.50% (50/101) more than 3 times a day. In the non-HE group, these results were 40.35% (23/57), 43.86% (25/57), and 15.79% (9/57), respectively (Figure 19 and Figure 20).

Additionally, the frequency of moisturizing the skin of the hands in respondents from the other study groups (before and during the pandemic) is presented in Appendix A of the Appendix A.

During the pandemic, moisturizing was significantly more frequent in the HE-study group compared to the non-HE group, HE-n/derm to non-HE, and HE-derm to non-HE (*p* < 0.001, *p* < 0.05, *p* < 0.001, respectively). There was no significant correlation between HE-n/derm and HE-derm with regard to the moisturizing of the skin of the hands in these study groups (Figure 21).

#### 3.2.8. Skin Infection

Before the pandemic period, respondents from the HE-study and non-HE groups reported bacterial, viral, and fungal infections of the skin of the hands. Among the respondents from the HE-study group, these were reported by 15/102 (14.71%) participants (nine bacterial, five viral, and one fungal infection), while in the non-HE group, these were reported by 4/68 (5.88%) (three bacterial and one viral). During the pandemic, these numbers decreased, five participants from the HE-study group reported one fungal superinfection, and four subjects declared mixed infections, fungal-bacterial (n = 2) and fungal-viral (n = 1). In the non-HE group, there was one person with fungal infection (Figure 22).

There was a significant decrease in the frequency of hand skin infections before and during the pandemic between: the HE-derm group before and the HE-derm group during the pandemic, and the HE group before the pandemic and the HE group during the pandemic (*p* < 0.05, *p* < 0.05, respectively) (Figure 23).

#### 3.2.9. Quality of Life

In the assessment of the quality of life (QoL) associated with the usage of hand disinfection, estimated using a visual analog scale, there was a significant decrease in QoL in the HE-study group compared to the non-HE group during the pandemic (*p* < 0.05). In the HE-study group, 67/102 respondents (65.69%) indicated answers 4 and 5, which suggested a significant reduction in quality of life, associated with hand disinfection, and in the non-HE group, only 18/68 persons (26.47%) perceived disinfection to be a nuisance (*p* < 0.05) (Figure 24).

Additionally, the assessment of quality of life associated with the hand disinfection in all four study groups during the pandemic is presented in Appendix A of the Appendix A.

During the pandemic, the quality of life of groups HE vs non-HE, HE-derm vs HE-n/derm, HE-n/derm vs non-HE, and HE-derm vs non-HE was reduced (*p* < 0.001, *p* < 0.001, *p* < 0.05, *p* < 0.001, respectively) (Figure 25).

## 4. Discussion

Recommendations of world and state organizations to limit the spread of the virus responsible for the development of the COVID-19 pandemic contributed to the improvement of hand skin hygiene. Due to easy access to disinfectants and their frequent use, the American Contact Dermatitis Society (ACDS) expects an increase in the incidence of allergic contact dermatitis and irritant dermatitis [4].

This survey aimed to investigate the prevalence of new hand skin symptoms related to skin disinfection during the COVID-19 pandemic, particularly in persons with a history of hand skin conditions. Furthermore, we intended to evaluate hand hygiene behaviors in relation to the pandemic.

Persons from the non-HE study group without previous history of any skin diseases, especially of the hands, and those from the HE-study group with hand skin dermatoses disinfected the skin with a similar daily frequency. We conclude that respondents from both groups disinfected their skin in comparable prevalence, thus, the results we obtained in further study sections could be compared due to the similarity of both groups. Additionally, Kendziora et al. showed that the frequency of hand washing and disinfecting the skin of the hands during the SARS-CoV2 pandemic increased compared to that prior to the pandemic [22].

Atopic dermatitis is one of the risk factors for the development of hand eczema [22]. The most frequent hand skin dermatose in the group of participants who were diagnosed by physician with hand skin eczema (HE-derm group) (58.93%) was hand AD and HE in the course of AD. From this group of respondents, nearly 50% observed new symptoms associated with disinfection, i.e., erythema, itchiness, scaling, pain, and fissures, that suggest the onset of hand eczema, and these outcomes correspond to the results obtained by other authors [23,24]. In the HE-study group, nearly 80% of respondents reported the occurrence of new symptoms. This demonstrates that distinguishing the HE-study group as including individuals who had not been diagnosed by a physician but had symptoms associated with HE greatly increased the number of respondents reporting new symptoms that had never been seen before. In order to accurately diagnose and help the patient determine the skin condition aggravating factors, doctors should more carefully examine the patient’s symptoms, especially those who visit the clinic for the first time. In particular, attention should be paid to the taking of a thorough history and determining whether the present symptoms are the result of a primary skin lesion observed by the patient, or perhaps an exacerbation of skin disease to which the patient previously had not paid attention. In modern medicine, it should be noted that patients tend to seek medical guidance via the Internet or telephone. In accordance, Singh et al. recently reported an increasing number of patients with HE diagnosed by telemedicine in India during the current pandemic [25]. This could simplify access to the physician, and therefore to make an appropriate diagnosis and initiate adequate treatment.

In the time of the pandemic, skin moisturizing was more often used for therapeutic rather than prophylactic purposes, due to the inclusion of emollients only when respondents began to experience new skin ailments [26]. According to a study from the outpatients’ clinic in Munich, patients with hand eczema applied emollients more frequently, which was further shown in our study [22]. During the pandemic, the use of emollients among our participants increased in both groups–in the HE-study group from 94% to 99%, and in the non-HE group from 79% to 83%,. Although the difference between the non-HE and HE respondents is not significant, the most important fact is that the daily frequency of emollient use increased in both groups. Due to the composition of emollients (humectants, fats, and oils) which replace depleted skin lipids and improve the function of the skin barrier, conducting an information campaign on the benefits of using emollients and the positive effect of moisturizing the skin on preventing the development of hand eczema among healthy people and reducing the number of exacerbations in patients with dermatological diseases should be considered as a future course of action [27]. It is important to mention that Symanzik et al. observed a rise in the frequency of emollient use among study participants, which contributed to improved skin condition at follow-up [15].

The group of respondents suffering from hand dermatoses seemed to be particularly prone to skin lesions at times of frequent disinfection. In the HE-study group, nearly 80% of respondents and 10% of the non-HE study group indicated new skin symptoms caused by the use of disinfectants. People with hand skin conditions more frequently reported the occurrence of new symptoms (three and more), the symptoms were also more severe (vesicles and pustules), and had never been observed by healthy respondents. New skin lesions reported in the HE group may additionally aggravate the course of their underlying diseases, causing therapeutic or diagnostic difficulties. Thus, patients with a dermatological history are at an increased risk of developing new skin symptoms during the pandemic and should be expected to present more often in dermatology practices [25]. Techasatian et al. showed that having previous underlying skin dermatose, i.e., atopic dermatitis, had a strong association with an increased risk of developing negative effects related to skin disinfection [28]. Exposure to detergents and soaps and frequent hand washing are well-known risk factors for exacerbation of underlying hand skin conditions [29]. In the HE-study group, over 60% reported more frequent exacerbations during the SARS-CoV-2 pandemic, and these outcomes correspond to the results obtained by Pourani et al. [18]. In addition, about 10% of the participants from the non-HE group also indicated that they were experiencing exacerbations caused by disinfection, however, due to the absence of primary hand skin diseases among these respondents, we believe that such responses were due to experiencing skin symptoms during disinfection and misunderstanding the survey question. Nevertheless, it is worth adding that the respondents from the HE and non-HE study group used disinfectants in the amounts recommended by the producers, hence, the observation of new symptoms and more frequent exacerbations should not result from an incorrect technique of disinfecting the skin of the hands. It is interesting to add that we observed that participants with hand skin dermatoses disinfected the skin of their hands less often than respondents without hand skin symptoms. It should be noted that the population with hand skin dermatoses is at higher risk of new symptoms appearing, hence, the usage of the appropriate amount of disinfectant and proper hygiene techniques (such as lukewarm-water hand washing, thorough hand drying, and moisturizing the skin after washing and during the day with fragrance-free products) is crucial to minimalize the negative disinfection effects [30].

Additionally, the application of a disinfectant caused symptoms which appeared shortly after the agents’ usage in both study groups. These symptoms were observed more frequently in persons from the HE-study group. Many years of research show that disinfection damages the normal hydrolipid barrier of the skin [30,31]. The additional exposure of such respondents to the intensive use of alcohol-based agents during the pandemic seems to predispose to new symptoms even more. Disinfectants reach the deeper layers of the epidermis and dermis, and this might explain the occurrence of pain and burning sensation in nearly 90% of respondents in the HE-study group immediately after application of the disinfectant.

Ong et al. and Haslund et al. observed that individuals suffering from hand skin diseases due to the disturbed protective layer of the skin, as well as immune processes, are at increased risk of microbiological infections, in particular bacterial (*S. aureus*) and viral [32,33]. In addition, respondents’ exposure to the frequent use of disinfectants may further damage the hydrolipid layer that protects against penetration of pathogenic microorganisms into the tissues. However, in our research, during the pandemic, an overall reduction in the number of microbial skin infections was observed in the HE and non-HE study groups. Possibly, two different results could be expected in this study section: increase in the number of infections, as well as its decrease. Due to the easy access to disinfecting agents, and intensified hand hygiene, the potentially pathogenic microorganisms are more promptly removed from the skin surface. This could potentially result in the reduction of skin infections. On the other hand, as a result of skin hydrolipid barrier damage, infections could appear more frequently. Interestingly, in the HE-study group, mixed infections with fungal-bacterial and fungal-viral origin appeared in respondents from this group during the pandemic. Further research is needed.

In our study, the appearance of new symptoms resulted in the usage of advanced methods of treatment (e.g., changed from topical to systemic treatment, increased the treatment dosage, or new drug added) only in persons from the HE-study group. This indicates that respondents from this study group experienced more severe negative effects of disinfection compared to the non-HE study group. Shah et al. recommended reducing the dosage of prednisolone or other prednisolone-sparing immunosuppressants to reduce the potential risk of SARS-CoV-2 infection in patients with AD [34]. On the other hand, considering the advantages and disadvantages of this recommendation, Pourani et al. investigated that reducing or stopping the immunosuppressant treatment in AD patients (in order to avoid COVID-19 infection) was related to an exacerbation of atopic dermatitis course in these patients [18].

The HE-study group visited a dermatologist more often during the pandemic, which is clearly related to the deterioration of their skin condition. Poorly treated skin lesions worsen the course of skin dermatoses [35]. We are confident that the respondents in our study perceived the need for seeking medical advice. Thus, we expect that the effects of an adequate therapy introduced by a specialist would be noticed faster and bring long-term positive change in skin condition. Pourani et al. mentioned that the lowest effective dose of immunosuppressants that controls the course of disease might reduce unnecessary visits for the monitoring of AD patients [18]. However, in order to determine the proper drug dosage, patients should firstly contact the physician.

The pandemic has had a profound impact on the functioning of all persons, affecting many aspects of their lives. The sanitary and epidemiological rigor and the exacerbations of skin lesions resulted in a significant reduction in the quality of life, especially in respondents with hand skin dermatoses. Over 65% of participants from the HE-study group, in the assessment of QoL related to disinfection, chose answers 4 and 5, which correlate with low quality of life during the pandemic. Comparitively, in the non-HE group, only 26% perceived disinfection to be a nuisance. This further proves the pandemic affected both physical and mental health.

Stress caused by a new situation and suddenly introduced changes in daily activity may itself be a factor exacerbating and triggering new skin diseases [36]. In addition, isolation and fear for the lives and well-being of relatives were inseparable parts of everyday pandemic life. Lewicka et al. reported that social COVID-19 exposure was associated with increased levels of stress and anxiety [37]. A review of the literature revealed that certain stressors were linked to a greater number of exacerbations and a worsening course of the disease in AD patients [38,39]. Stress is a well-known trigger factor of AD and disrupts the proper functioning of the epidermal barrier in mechanism involving stress-induced secretion of endogenous glucocorticoids [40,41]. This might lead to damage of the skin barrier.

It is important to analyze the current situation and probably expect the emergence of a new group of patients, or more patients in general who need the advice of a dermatologist. It is also worth remembering the patients’ mental health as knowing how to control everyday stress may turn out to have a beneficial effect in the prognosis of skin diseases [42,43].

This study is limited due to its online method of surveying, as it was based on the personal perspective of respondents. As authors of this research, we could not check the actual condition of the respondents’ hand skin. The survey was not available to individuals without access to the Internet or social media. Most respondents were women, who were more likely to take part in online surveys on dermatological topics.

Studies on the influence of particular types of disinfecting agents on developing skin lesions in both groups are required to identify which factors may exacerbate the hand skin lesions and induct new symptoms over the skin of the hands. Further investigations are needed to improve the treatment guidelines targeted to HE and a patient’s needs, namely personalized medicine.

## 5. Conclusions

The COVID-19 pandemic affected patients’ and healthy individuals’ lives. This study revealed a meaningful increase in the prevalence of hand skin symptoms and deterioration of the skin condition. Reduction of the quality of life of respondents in the HE-study group was observed. More frequent skin lesion exacerbations and reduced possibility of obtaining remission were reported. In fact, as disinfectants interrupt the hydrolipid barrier, education on appropriate disinfection techniques and skincare, as well as early dermatological intervention, might allow us to limit the development of hand skin diseases.

## Figures and Tables

**Figure 1 jcm-12-00195-f001:**
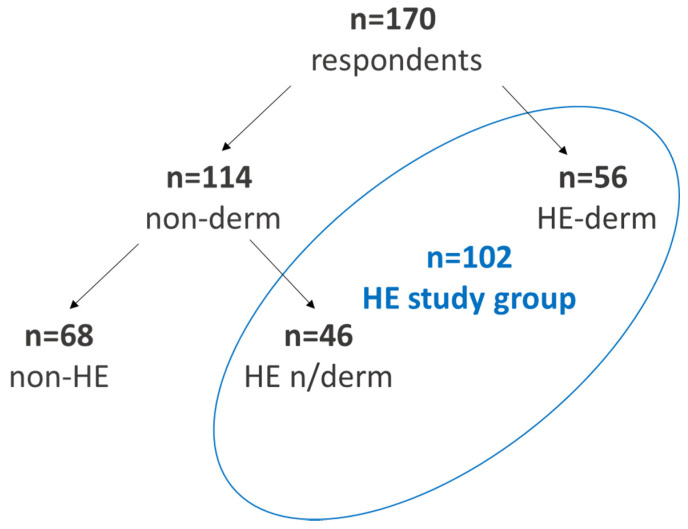
The division of study groups.

**Figure 2 jcm-12-00195-f002:**
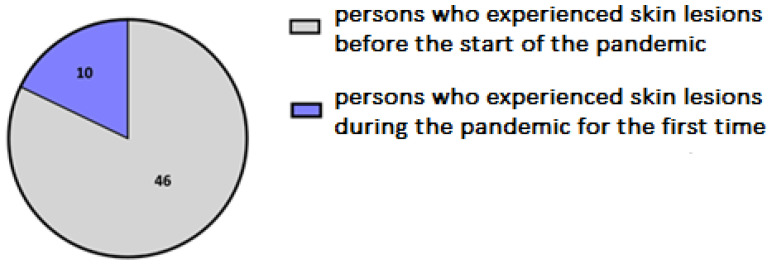
Division of the HE-derm group according to the time of diagnosis of the hand skin disease.

**Figure 3 jcm-12-00195-f003:**
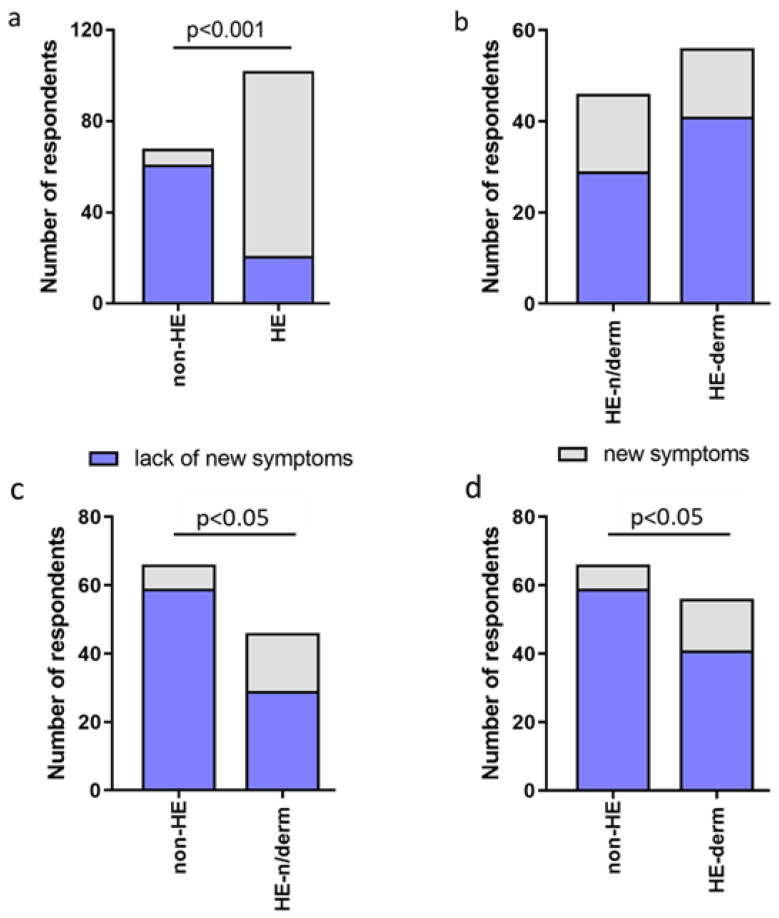
Comparison of occurrence of new symptoms in study groups: (**a**) non-HE vs. HE; (**b**) HE-n/derm vs. HE-derm; (**c**) non-HE vs. HE-n/derm; (**d**) non-HE vs. HE-derm.

**Figure 4 jcm-12-00195-f004:**
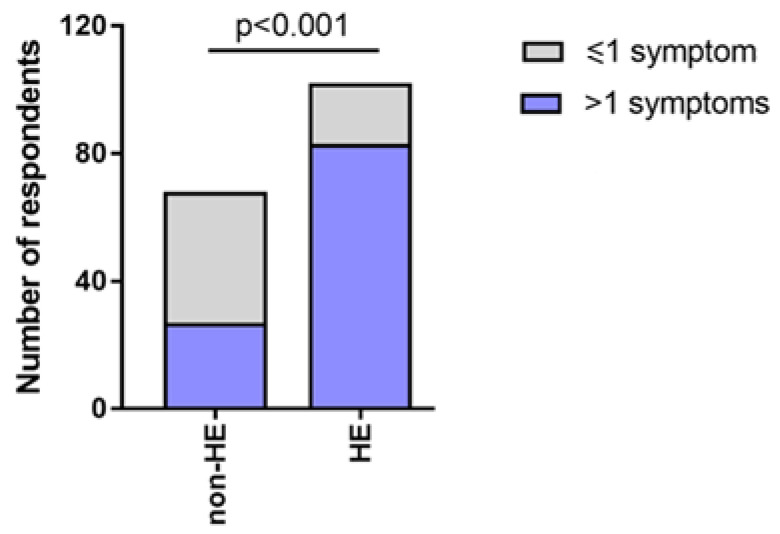
Number of symptoms reported by the HE-study group and the non-HE group immediately after application of the disinfectant.

**Figure 5 jcm-12-00195-f005:**
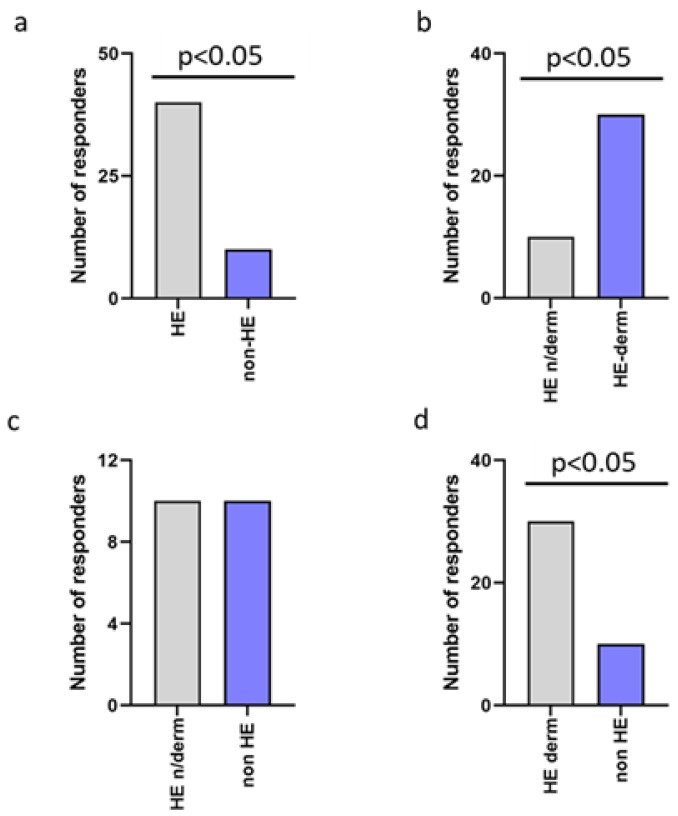
Comparison of the frequency of pruritis reported in the study groups: (**a**) non-HE vs. HE; (**b**) HE-n/derm vs. HE-derm; (**c**) non-HE vs. HE-n/derm; (**d**) non-HE vs. HE-derm.

**Figure 6 jcm-12-00195-f006:**
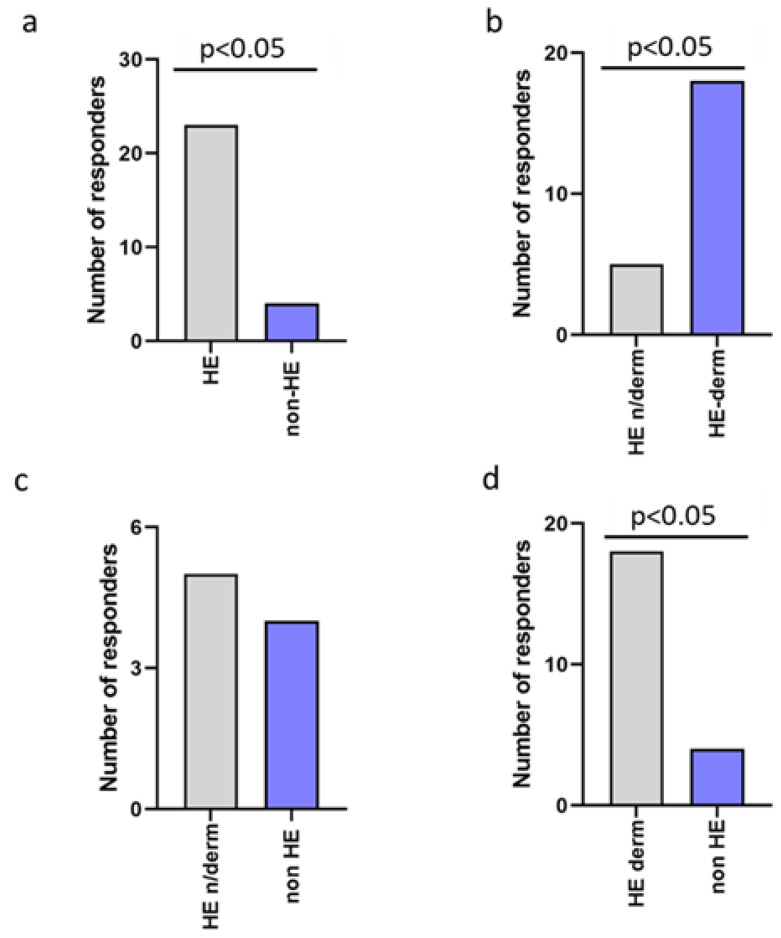
Comparison of the frequency of increased skin warmth reported in the study groups: (**a**) non-HE vs. HE; (**b**) HE-n/derm vs. HE-derm; (**c**) non-HE vs. HE n/derm; (**d**) non-HE vs. HE-derm.

**Figure 7 jcm-12-00195-f007:**
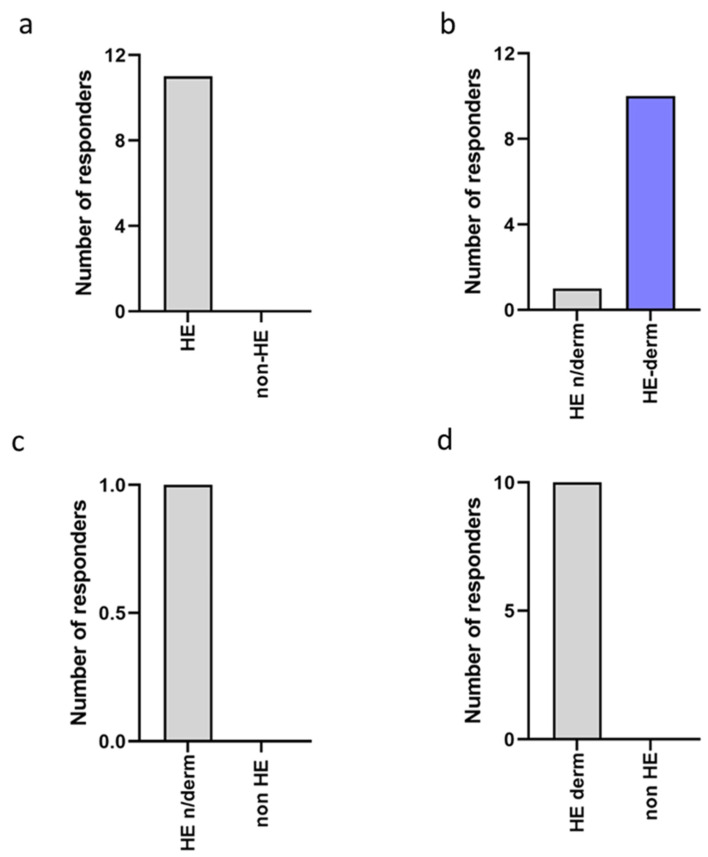
Comparison of the frequency of edema reported in the study groups: (**a**) non-HE vs. HE; (**b**) HE n/derm vs. HE-derm; (**c**) non-HE vs. HE n/derm; (**d**) non-HE vs. HE-derm.

**Figure 8 jcm-12-00195-f008:**
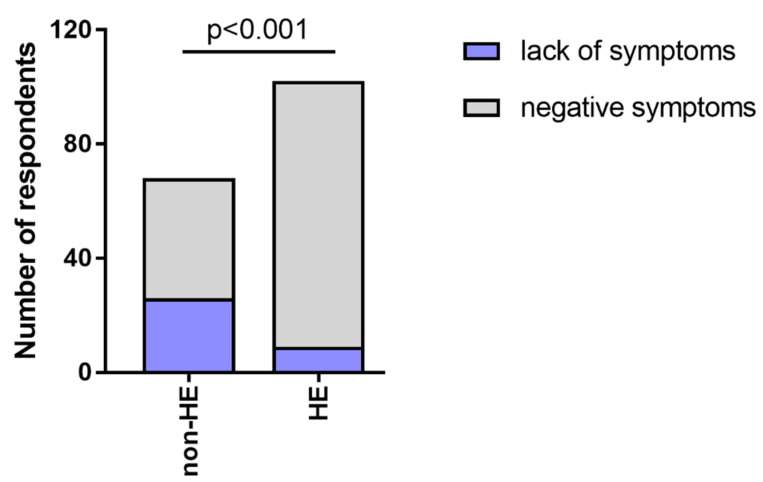
Negative effects and lack of negative effects during the application of the antiseptic experienced in both groups (HE-study group vs non-HE).

**Figure 9 jcm-12-00195-f009:**
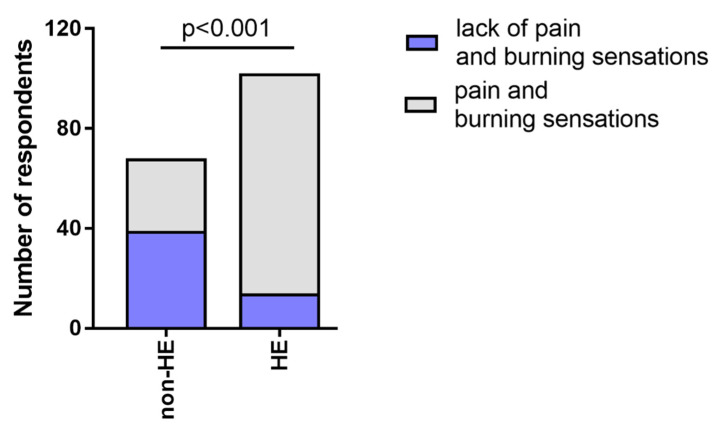
The frequency of pain and burning sensations after using disinfectants in both groups.

**Figure 10 jcm-12-00195-f010:**
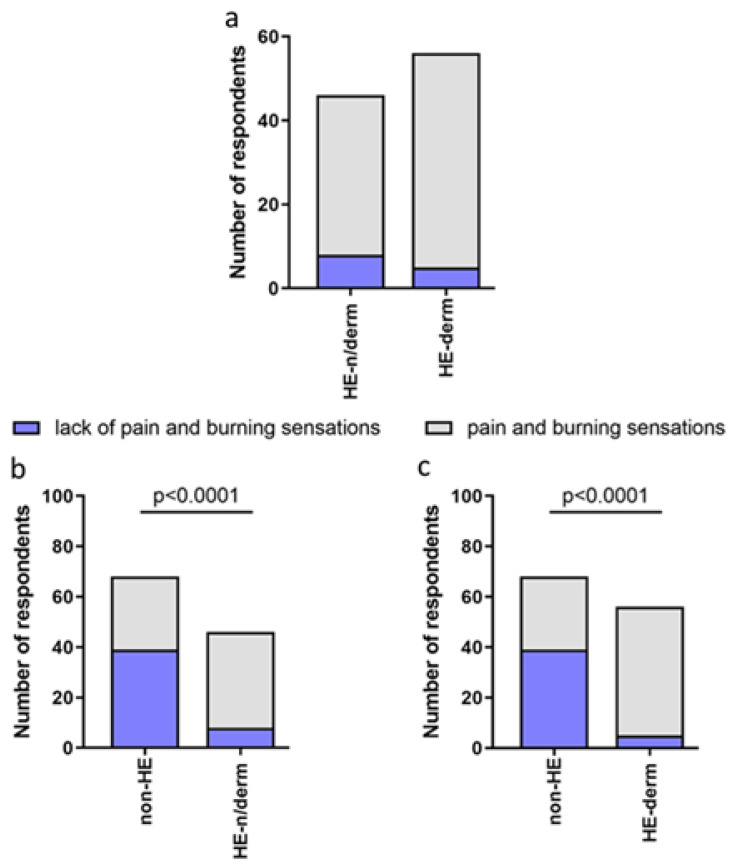
Comparison of the frequency of pain and burning sensations after using disinfectants in study groups: (**a**) HE-n/derm vs. HE-derm; (**b**) non-HE vs. HE-n/derm; (**c**) non-HE vs. HE-derm.

**Figure 11 jcm-12-00195-f011:**
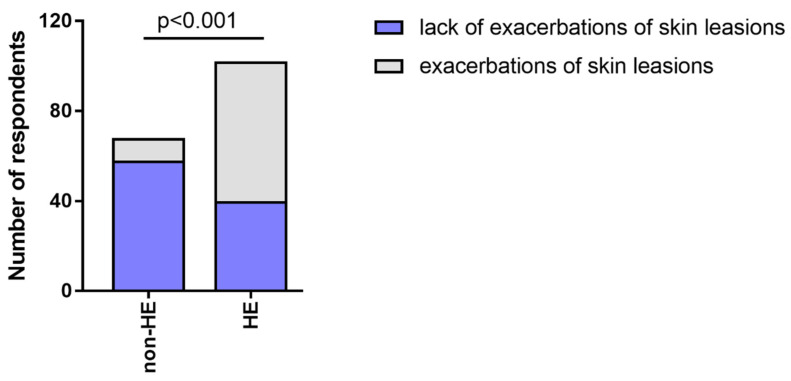
The frequency of exacerbations of skin lesions during the pandemic in both groups (HE-study group and non-HE).

**Figure 12 jcm-12-00195-f012:**
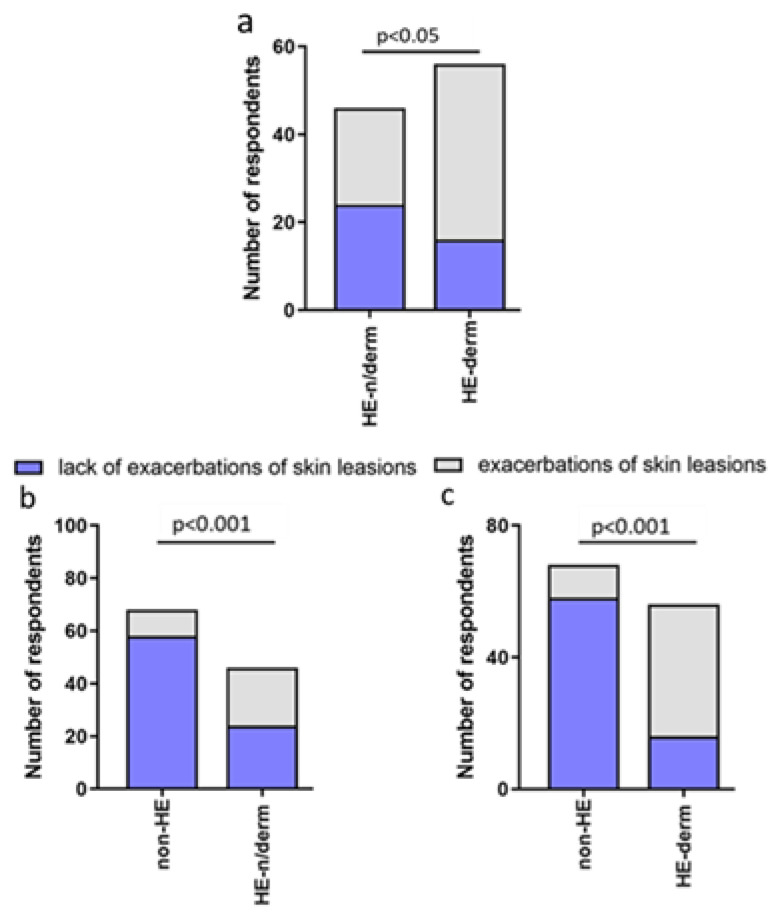
Comparison of the frequency of exacerbations of skin lesions during the pandemic in study groups: (**a**) HE-n/derm vs. HE-derm; (**b**) non-HE vs. HE-n/derm; (**c**) non-HE vs. HE-derm.

**Figure 13 jcm-12-00195-f013:**
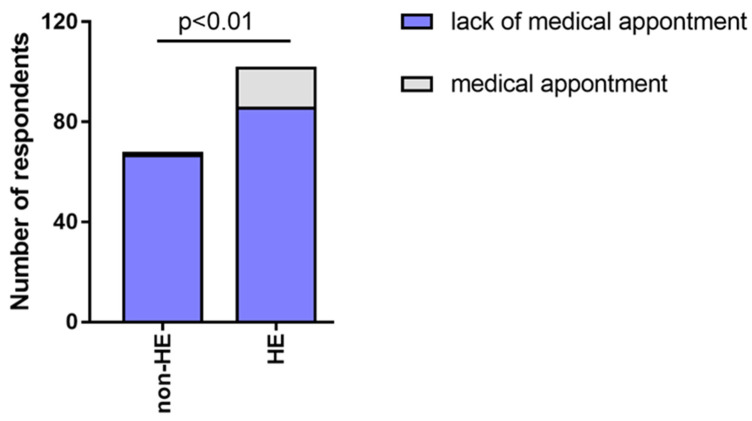
The frequency of medical appointment during pandemic in both groups (HE-study group and non-HE).

**Figure 14 jcm-12-00195-f014:**
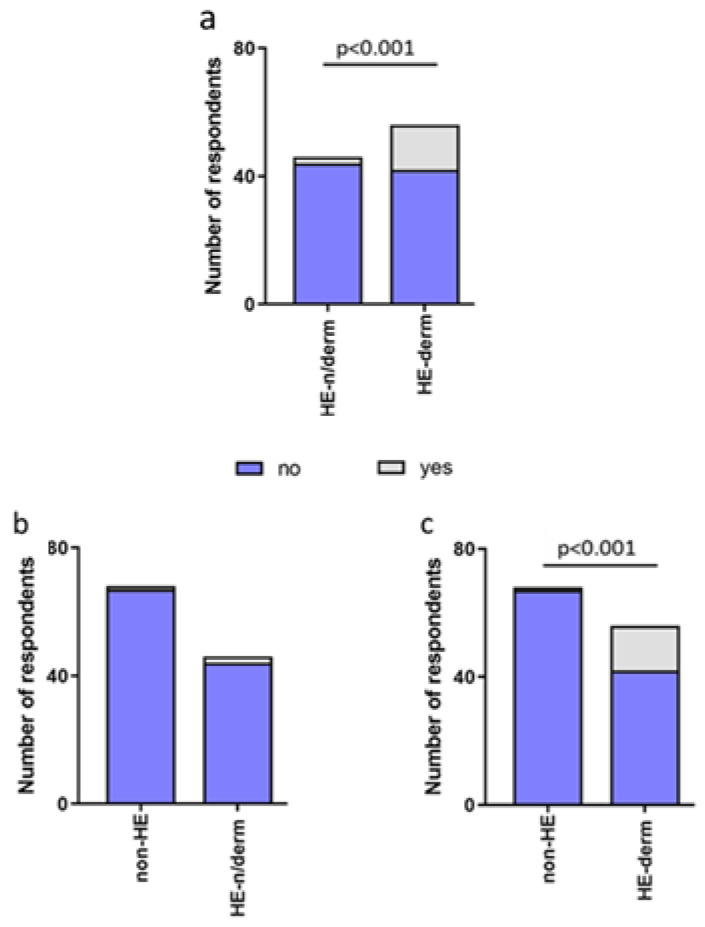
Comparison of the frequency of medical appointments during pandemic in study groups: (**a**) HE-n/derm vs. HE-derm; (**b**) non-HE vs. HE-n/derm; (**c**) non-HE vs. HE-derm.

**Figure 15 jcm-12-00195-f015:**
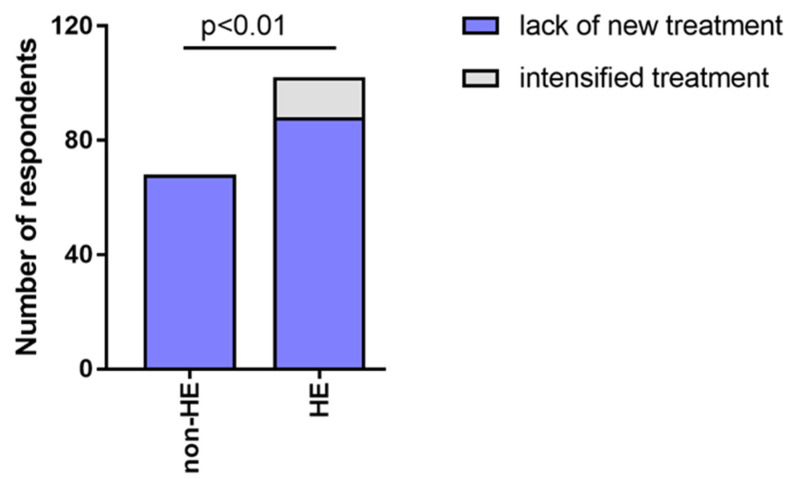
Changes in treatment in the HE-study group and non-HE group during pandemic.

**Figure 16 jcm-12-00195-f016:**
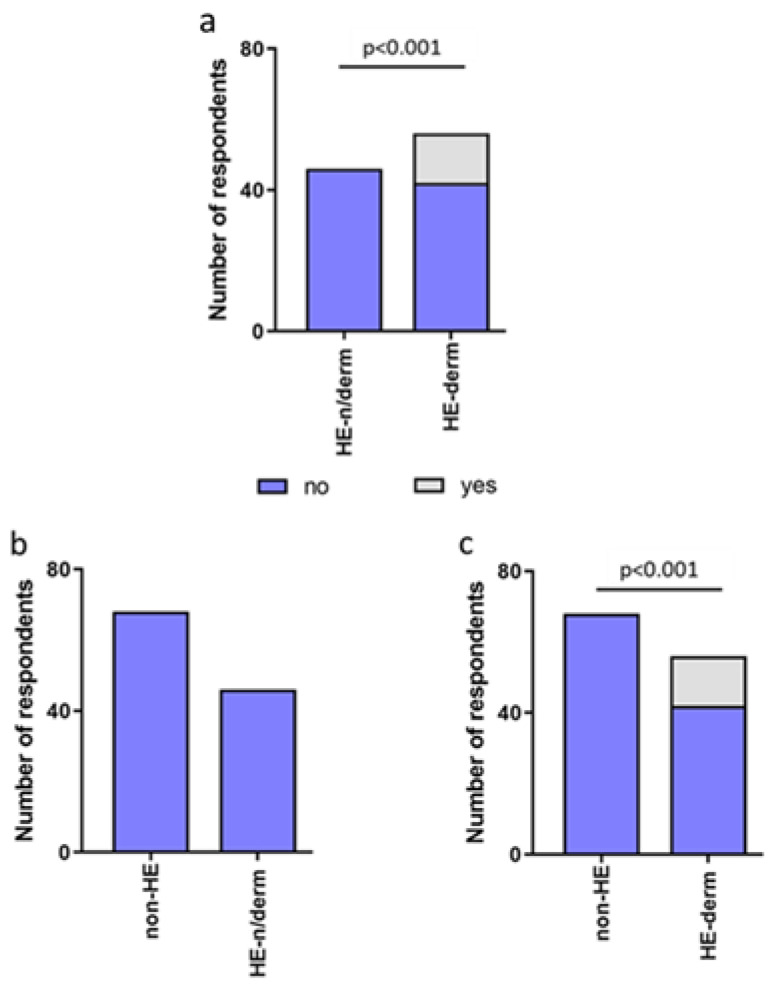
Changes in treatment in the study groups during the pandemic: (**a**) HE-n/derm vs. HE-derm; (**b**) non-HE vs. HE-n/derm; (**c**) non-HE vs. HE-derm.

**Figure 17 jcm-12-00195-f017:**
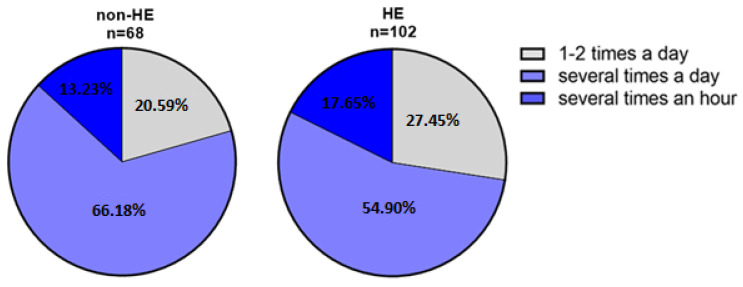
Percentage distribution of hand disinfection frequency in both groups during the pandemic.

**Figure 18 jcm-12-00195-f018:**
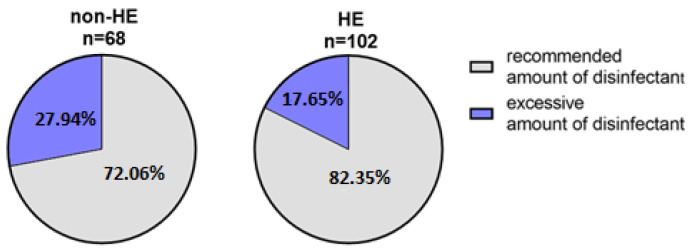
Amount of disinfectant applied by respondents.

**Figure 19 jcm-12-00195-f019:**
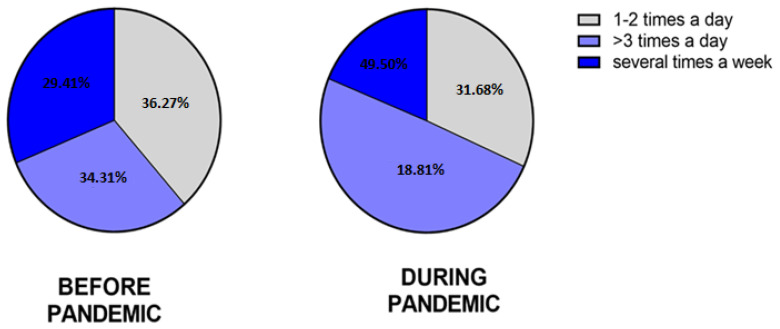
The frequency of moisturizing the skin of the hands in the HE group of respondents before and during the pandemic.

**Figure 20 jcm-12-00195-f020:**
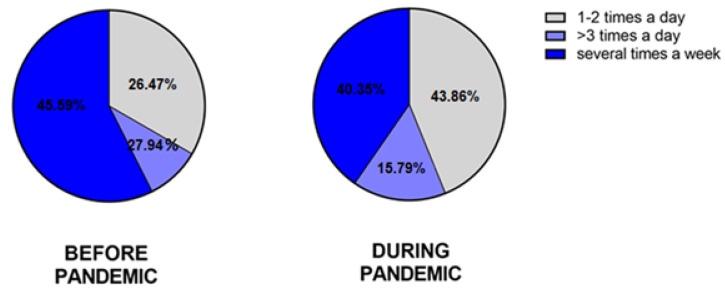
The frequency of moisturizing the skin of the hands in the non-HE group of respondents before and during the pandemic.

**Figure 21 jcm-12-00195-f021:**
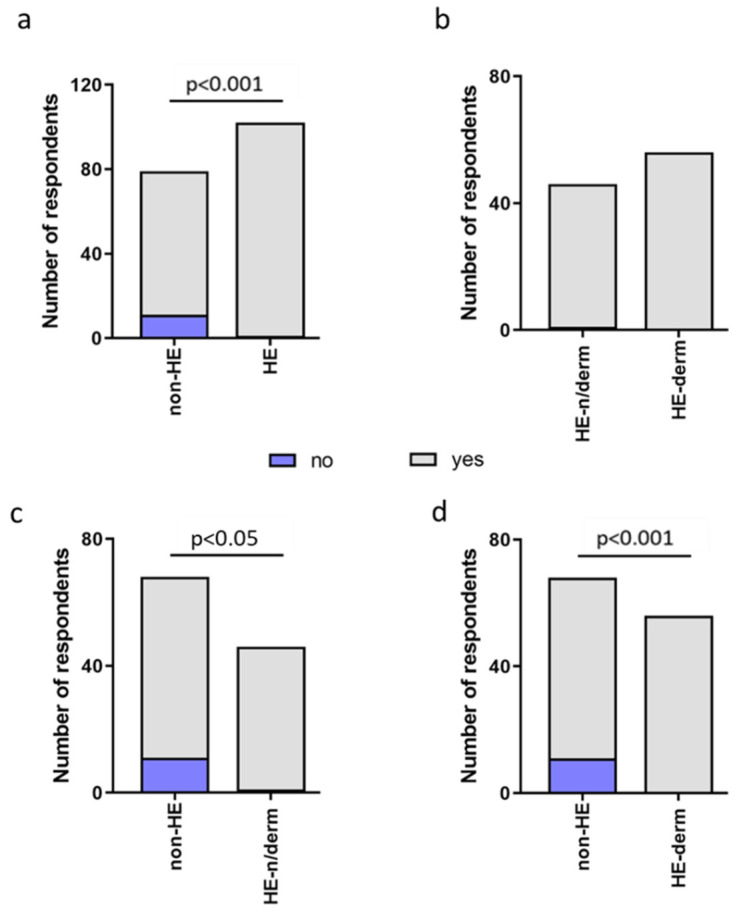
Comparison of moisturizing of the skin of the hands in study groups during pandemic: (**a**) non-HE vs. HE; (**b**) HE n/derm vs. HE-derm; (**c**) non-HE vs. HE n/derm; (**d**) non-HE vs. HE-derm.

**Figure 22 jcm-12-00195-f022:**
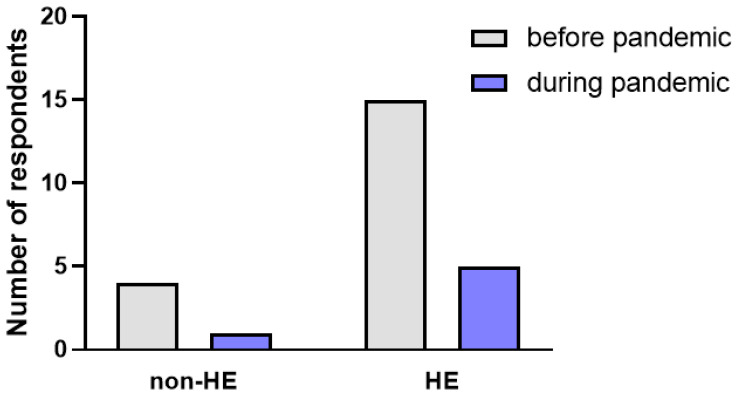
Comparison of the prevalence of skin infections in both groups (non-derm and HE-derm) before and during the pandemic.

**Figure 23 jcm-12-00195-f023:**
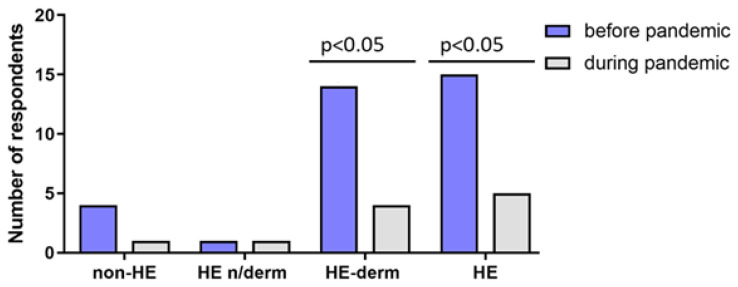
Comparison of the prevalence of skin infections in the study before and during the pandemic.

**Figure 24 jcm-12-00195-f024:**
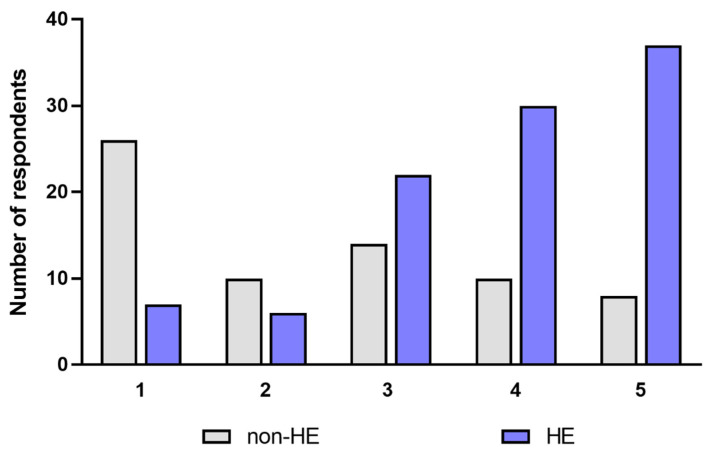
Assessment of the quality of life associated with hand disinfection in both groups (HE-study group and non-HE) during the pandemic. 1-disinfection had no negative effect on the quality of life, 5–disinfection had a significant impact by lowering quality of life.

**Figure 25 jcm-12-00195-f025:**
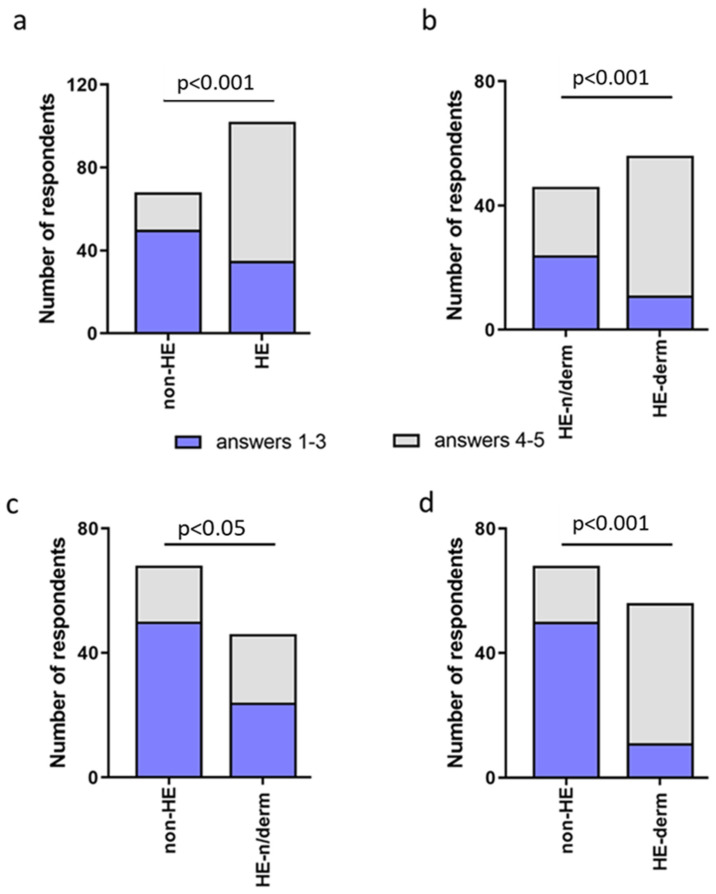
The number of respondents in study groups who chose the answers 1–3 and 4–5 in the assessment of quality of life: (**a**) non-HE vs. HE; (**b**) HE n/derm vs. HE-derm; (**c**) non-HE vs. HE n/derm; (**d**) non-HE vs. HE-derm.

**Table 1 jcm-12-00195-t001:** Demographic characteristics of the study group.

Characteristics	N (%)
**Gender**	
Female	142 (83.53%)
Male	28 (16.47%)
**Age (years)**	
18–24	89 (52.35%)
25–36	70 (41.18%)
36–45	5 (2.94%)
>45	6 (3.53%)
**Education**	
Primary	2 (1.18%)
Secondary	84 (49.41%)
Higher	84 (49.41%)
**Residence**	
Rural	21 (12.35%)
City to 250.000 inhabitants	54 (31.77%)
City with more than 250.000 inhabitants	95 (55.88%)

**Table 2 jcm-12-00195-t002:** Three most frequent hand skin dermatoses in HE-derm group.

Hand Skin Dermatoses	N (%)
Hand eczema	56 (100.00%)
Atopic dermatitis	33 (58.93%)
Hand and nail psoriasis	8 (14.29%)

**Table 3 jcm-12-00195-t003:** The frequency of new symptoms reported by respondents in both groups (HE-study group vs. non-HE).

	HE-Study Group, N (%)	Non-HE, N (%)
**Number of respondents**	81 (79.41%)	7 (10.29%)
**New symptom**		
Dryness	62 (60.78%)	3 (4.41%)
Roughness	53 (51.96%)	2 (2.94%)
Redness	40 (39.22%)	0
Vesicles filled with serum fluid	7 (6.86%)	0
Pus-filled pustules	5 (4.90%)	0

## Data Availability

Not applicable.

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
