# Peer review of "Questionnaire-Based Study Evaluating the Hand Hygiene Practices and the Impact of Disinfection in the COVID-19 Pandemic on Hand Skin Conditions in Poland"

_jcm, 2022, doi:10.3390/jcm12010195_

Round 1

Reviewer 1 Report

The reviewed manuscript illustrates a quite interesting study with a high potential in its results. However, the massive editing of both content and the form of presentation is required before the re-evaluation and potential publication.

Here are my biggest concerns:

1. The Introduction section is way too short and it does not bring satisfactory background for the future readers. I would recommend expanding it with several issues, including:

a) the impact and potential sequelae of hand hygiene mainly in health-care proffesionals BEFORE the pandemic (e.g. Zhang et al. doi: 10.1111/ajd.12672; Gasparini et al. doi: 10.23736/S0392-0488.19.06220-5; Hamnerius et al. doi 10.1111/cod.13042) and during it (Loh et al. doi 10.1111/cod.14133);

b) the general epidemiology of hand eczema regardless of pandemic (Mora-Fernandez et al. doi 10.1016/j.ad.2021.10.002; Quaade et al. doi 10.1111/cod.13804)

c) an outlook of treatment possibilities administered with or without contact with a dermatologist (Symanzik et al. doi 10.1111/cod.14206; Thyssen et al. 10.1111/cod.14035)

2. The Materials and methods section does not contain the most crucial part, that is the construction of the survey. A potential reader would be interested in the final questionnaire and motivation of authors to formulate their questions in the chosen manner. There is also an inaccuracy regarding the informed consent. How is this possible that the authors obtained a written ICF from each participant if the survey was utilized as an online questionnaire only? Please clarify.

3. The Results section is built on a basis under the working title "let us analyze everything with everything and show them what we got". There is no clinical filter in the current version which is mandatory to describe and comment the statistical analysis properly. For instance: one does not have to be a doctor or even peform the statistics to conclude that:

a) hand eczema diagnosis is mandatory to be included in HE-derm group (line 159);

b) patients with confirmed HE diagnosis had to seek medical advice more frequently than healthy controls (lines 284-286);

c) participants from non-HE group did not report the significant impact of their quality of life (lines 420-421).

It is quite disturbing that the authors make so many overstatements on the statistical significance of the presented comparisons, given that the whole outcome of the study is limited to descriptive analysis only. A large proportion of p values lower than 0.05 appears to be false positive and it requires meticulous evaluation by experienced statistician before resubmission. For instance, it is so incorrect to draw a conclusion that dryness or roughness are significantly more frequent in HE study group than in non-HE patients if the latter group consists of 7 patients only (Table 3). Did the authors analyze standard deviations or confidence intervals?

I am very interested in an issue partially presented in Figures 17 and 19. Can the authors expand this subsection with elaboration on quantities how many patients altered their pre-pandemic habits?

4. The Discussion section does not focus accurately on the reported results and should be rewritten with adjusted accentuation of the most important results from the authors' standpoint. The commentary should never be one-sided. For instance, why the decrease of microbial skin infections (lines 536-537) is considered as an unexpected result (lines 541-542)? Isn't it the overall objective of hand hygiene procedures? Please elaborate on this issue. By the way, stress-trigerred dermatitis is an important and fascinating topic but should not be placed in the middle of the paragraph about disinfection because it only makes a mess in the already difficult section.

I am also deeply concerned that Discussion does not contain at least a sentence about the limitations of the study. Pointing drawbacks and future ways of improvement would show the perspective of the experienced researchers involved in the study team.

Quality of presentation merits improvement as well. Table 1 provided the same information as lines 150-155 and there is no need of such reiteration. If the authors implement any kind of abbreviation, they should be consequent and use it in the rest of the text (e.g. line 165 contains "hand skin eczema" again). Every commentary should be exact to ease the interpretation for the future readers (e.g. line 301 - what does "more intensive therapy" stand for?).

In the end, the whole manuscipt requires heavy language editing. There are many sentences written in English but with Polish syntax (e.g. lines 458-459). In a few cases the dictionary did not work properly (e.g. line 100 "proprietary" instead of "proper/appropriate"). What does "statistically significance difference" mean (e.g. line 198)? There are also numerous mistyping errors (e.g. lines 51-52 "detoriation" instead of "deterioration"; lines 55 doubling "our").

I am looking forward to review the improved version of this manuscript.

Author Response

RESPONSE TO REVIEWER 1:

The authors appreciate the thorough review and constructive comments. We have carefully assessed each of the comments and criticisms and we made changes to the manuscript accordingly. We believe that the revised manuscript has improved through this review process. We hope that the Editor and the Reviewers will agree with us. All changes to the manuscript after the Reviewer's suggestions are written in red.

We have provided a detailed response to Reviewer 1:

The reviewed manuscript illustrates a quite interesting study with a high potential in its results. However, the massive editing of both content and the form of presentation is required before the re-evaluation and potential publication.

Here are my biggest concerns:

  1. The Introduction section is way too short and it does not bring satisfactory background for the future readers. I would recommend expanding it with several issues, including:
  2. a) the impact and potential sequelae of hand hygiene mainly in health-care proffesionals BEFORE the pandemic (e.g. Zhang et al. doi: 10.1111/ajd.12672; Gasparini et al. doi: 10.23736/S0392-0488.19.06220-5; Hamnerius et al. doi 10.1111/cod.13042) and during it (Loh et al. doi 10.1111/cod.14133);
  3. b) the general epidemiology of hand eczema regardless of pandemic (Mora-Fernandez et al. doi 10.1016/j.ad.2021.10.002; Quaade et al. doi 10.1111/cod.13804)
  4. c) an outlook of treatment possibilities administered with or without contact with a dermatologist (Symanzik et al. doi 10.1111/cod.14206; Thyssen et al. 10.1111/cod.14035)

    We thank the Reviewer for this comment. We have added to our manuscript all of the studies listed above. We agree with the Reviewer that a thorough description of the impact and potential sequelae of hand hygiene in healthcare professionals before and during the pandemic, as well as the general epidemiology of hand eczema, and an outlook of treatment possibilities were missing in the introduction. We appreciate the help in pointing out the interesting literature. We are pleased to be able to supplement our manuscript with such valuable data.
  5. The Materials and methods section does not contain the most crucial part, that is the construction of the survey. A potential reader would be interested in the final questionnaire and motivation of authors to formulate their questions in the chosen manner. There is also an inaccuracy regarding the informed consent. How is this possible that the authors obtained a written ICF from each participant if the survey was utilized as an online questionnaire only? Please clarify.

    We appreciate this suggestion. We have created Table S1 showing the questions included in our survey, and added it to the Supplementary Material. Due to the fact, that the survey consisted of many questions, we decided to put the new Table S1 as part of Supplementary Material, as the long table would appear cumbersome to read in the main manuscript text. We truly believe that this will allow future readers to study the content of the survey.

    During the literature review on subjects related to our manuscript, we came to the conclusion that it would be interesting to create a study that assesses the multi-problem state of Polish patients suffering from Hand Eczema, summarizes and shows how behaviors regarding disinfection and moisturizing the skin of the hands differ between distinguished groups.

We wanted to create a study describing many factors and behaviors that could affect the condition of the skin of the hands. Due to the introduced restrictions, we, unfortunately, did not have the opportunity to meet with persons who were experiencing new disinfection-related symptoms, hence an online survey seemed to be the safest form of communication at the time.

We thank the Reviewer for this comment. There was an editorial error while writing the manuscript. We agree that it was impossible to collect written ICF, as this was an anonymous online survey. We want to, once again, add as clarification, that prior to participating in the survey, individuals were informed about the conducted research and data collection. Participation in the study was voluntary. Participants could withdraw their consent at any time during the survey. All respondents provided informed consent. The study was approved by the local Bioethics Committee. We have made changes to the main manuscript text.

  1. The Results section is built on a basis under the working title "let us analyze everything with everything and show them what we got". There is no clinical filter in the current version which is mandatory to describe and comment the statistical analysis properly. For instance: one does not have to be a doctor or even peform the statistics to conclude that:
  2. a) hand eczema diagnosis is mandatory to be included in HE-derm group (line 159);

    We thank the Reviewer for this comment. We want to clarify; this is the description of the persons from the HE-derm group. All respondents from this study group had a diagnosis of hand eczema (HE). The respondents were diagnosed by a dermatologist. This is the reason the study group was named HE-derm. This information is obligatory to understand the division of the study groups and must remain in the description of the study population. This is the explanation of the division of the study population.
  3. b) patients with confirmed HE diagnosis had to seek medical advice more frequently than healthy controls (lines 284-286);
  4. c) participants from non-HE group did not report the significant impact of their quality of life (lines 420-421).

We thank the Reviewer for the comment. The research was directed not only to dermatologists, who are specialists in skin diseases and can predict the effects of frequent disinfection of the skin of the hands. As authors, we wanted our study to be accessible to doctors of various specialties, e.g., general practitioners, as well as the respondents themselves. We do not find our results to be of no merit. The data we collected is not obvious to all future readers. We agree that the 2 examples indicated by the Reviewer above are general, but we found them interesting to mention. In our opinion it is important that respondents from HE study group found the disinfection to be a nuisance, hence, on contrary, the results obtained by non-HE study group should be shown. Therefore, without these results, we would not be able to compare these two groups.
We agree that it's not necessary to be a doctor to conclude, but a researcher's role is to confirm a hypothesis based on gathered data. Additionally, studies regarding seemingly obviously true or false hypotheses often lead to unexpected results. General opinions should be documented with appropriate scientific research to build the basis of "evidence-based medicine".
We would like to emphasize that the data we obtained corresponds with the outcomes of many other authors (e.g., DoÄŸan EI, Birgül ÖK. New-onset or Exacerbated Occupational Hand Eczema among Healthcare Workers During the COVID-19 Pandemic: A Growing Health Problem. Acta Dermatovenerol Croat. 2021, Apr;291(1):21-29. PMID: 34477059; Babino G, Argenziano G, Balato A. Impact in Contact Dermatitis during and after SARS-CoV2 Pandemic. Curr Treat Options Allergy. 2022, 9(1):19-26. doi: 10.1007/s40521-022-00298-2; 14. Pourani MR, Ganji R, Dashti T, Dadkhahfar S, Gheisari M, Abdollahimajd F, Dadras MS. Impact of COVID-19 Pandemic on Patients with Atopic Dermatitis. Actas Dermosifiliogr. 2021, Sep 20:S0001-7310(21)00326-4. doi: 10.1016/j.ad.2021.08.004.), hence we consider these data to be relevant.

Referring to the Reviewer’s statement There is no clinical filter, which we consider very harmful, we would like to explain that our main goal was to compare the HE and non-HE groups, which we indicated by highlighting the graphs comparing these two groups as separate. The other comparisons between the groups were placed below descriptions of HE vs non-HE as additional information. However, we decided not to include these graphs in Supplementary Materials due to the fact the data was statistically significant many times. Additionally, we have not encountered such comparisons in any research we have studied, and we considered this a valuable element that gives the opportunity to assess the impact of disinfection on other groups.

It is quite disturbing that the authors make so many overstatements on the statistical significance of the presented comparisons, given that the whole outcome of the study is limited to descriptive analysis only. A large proportion of p values lower than 0.05 appears to be false positive and it requires meticulous evaluation by experienced statistician before resubmission. For instance, it is so incorrect to draw a conclusion that dryness or roughness are significantly more frequent in HE study group than in non-HE patients if the latter group consists of 7 patients only (Table 3). Did the authors analyze standard deviations or confidence intervals?

We thank the Reviewer for this comment. We have re-analyzed the data shown in Table 3 and presented the results more appropriately.
All results were prepared by an experienced statistician. They have been double-checked. We are confident in the correctness of the data presented.
In our studies relationships between two variables were analyzed using the chi-square independence test with 95% confidence intervals (CI). The Chi-square statistic is a non-parametric tool designed to analyze group differences when the dependent variable is measured at a nominal level. Like all non-parametric statistics, the Chi-square is robust with respect to the distribution of the data. Specifically, it does not require equality of variances among the study groups or homoscedasticity in the data. The most commonly used confidence interval is 95%. It indicates that the estimated range has a 95% chance of containing the true value. In our previous study, we also used a chi-square independence test with 95% confidence intervals (CI) (doi: 10.1007/s13555-021-00662-1, doi: 10.1007/s13555-022-00796-w).

I am very interested in an issue partially presented in Figures 17 and 19. Can the authors expand this subsection with elaboration on quantities how many patients altered their pre-pandemic habits?

We thank the Reviewer for this comment. Our survey was not directed only to respondents who had contact with disinfecting agents before the pandemic (e.g., medical professions, food service, etc.), but to the general population, in particular to persons with hand skin dermatoses. Our survey did not include the question on the frequency of disinfecting the skin of the hands prior to the pandemic outbreak, as we assumed that these agents became easily available in the public shortly after the restrictions were announced. Because of this, we are not in a position to provide the Reviewer with these pre-pandemic data.

Referring to the Reviewer's question about the numerical values for the frequency of hand skin moisturization before and during the pandemic, we would like to indicate that the answer to this question was, and still is, found between lines 381 and 390 of the main manuscript text. The missing data was added.

We thank the Reviewer for indicating the importance of the data collected. We consider them extremely valuable, especially during the pandemic.

  1. The Discussion section does not focus accurately on the reported results and should be rewritten with adjusted accentuation of the most important results from the authors' standpoint. The commentary should never be one-sided. For instance, why the decrease of microbial skin infections (lines 536-537) is considered as an unexpected result (lines 541-542)? Isn't it the overall objective of hand hygiene procedures? Please elaborate on this issue. By the way, stress-trigerred dermatitis is an important and fascinating topic but should not be placed in the middle of the paragraph about disinfection because it only makes a mess in the already difficult section.

    As authors, we have been searching for answers to the multi-problem state of patients suffering from Hand Eczema during the COVID-19 pandemic (e.g., the symptoms, exacerbations, treatment, and quality of life). We believe the results we presented in the discussion are relevant to both, daily and medical practice. In addition, our results correlate with those of many other authors cited in our study. Our research presents many aspects of the Polish HE patient population. It gives a comprehensive view of the problems affecting the local population. We realize that the data we collected could be used for more than one publication, but we decided that a study summarizing it all would be much more valuable.

    We thank the Reviewer for this comment. We agree with the Reviewer’s opinion. We should also have considered the possibility of reducing skin infections due to increased hand hygiene during the pandemic. The issue brought by the Reviewer was included in the discussion and can be found between lines 553-558.
    We based our primary assessment on the data provided by other studies (e.g., Zhou NY, Yang L, Dong LY, Li Y, An XJ, Yang J, Yang L, Huang CZ, Tao J. Prevention and Treatment of Skin Damage Caused by Personal Protective Equipment: Experience of the First-Line Clinicians Treating 2019-nCoV Infection. Int J Dermatol Venereol. 2020 Mar 13:10.1097/JD9.0000000000000085. doi: 10.1097/JD9.0000000000000085. Epub ahead of print. PMID: 34192087; PMCID: PMC7147274.; Aiello AE, Coulborn RM, Perez V, Larson EL. Effect of hand hygiene on infectious disease risk in the community setting: a meta-analysis. Am J Public Health. 2008 Aug;98(8):1372-81. doi: 10.2105/AJPH.2007.124610. Epub 2008 Jun 12. PMID: 18556606; PMCID: PMC2446461.; Pendlebury GA, Oro P, Haynes W, Merideth D, Bartling S, Bongiorno MA. The Impact of COVID-19 Pandemic on Dermatological Conditions: A Novel, Comprehensive Review. Dermatopathology (Basel). 2022 Jun 29;9(3):212-243. doi: 10.3390/dermatopathology9030027. PMID: 35892480; PMCID: PMC9326733.).

We thank the Reviewer for the valuable comment regarding stress-triggered dermatitis. In accordance, we have moved this part of the discussion to the quality-of-life section. We hope this is the appropriate place to discuss the impact of stress on the quality of life as well as hand skin condition.

I am also deeply concerned that Discussion does not contain at least a sentence about the limitations of the study. Pointing drawbacks and future ways of improvement would show the perspective of the experienced researchers involved in the study team.

We thank the Reviewer for this suggestion. We agree with the Reviewer's opinion, this was our oversight. Limitations have been added to the Discussion.

Quality of presentation merits improvement as well. Table 1 provided the same information as lines 150-155 and there is no need of such reiteration.

We thank the Reviewer for this comment. The description has been removed, and the data is presented only in Table 1.

If the authors implement any kind of abbreviation, they should be consequent and use it in the rest of the text (e.g. line 165 contains "hand skin eczema" again).

We thank the Reviewer for this comment. The phrases hand eczema and HE are used interchangeably in our manuscript. This allows us to avoid repetition which could affect the quality of the text.

 Every commentary should be exact to ease the interpretation for the future readers (e.g. line 301 - what does "more intensive therapy" stand for?).

We thank the Reviewer for this comment. The explanation has been added in parentheses.

In the end, the whole manuscipt requires heavy language editing. There are many sentences written in English but with Polish syntax (e.g. lines 458-459). In a few cases the dictionary did not work properly (e.g. line 100 "proprietary" instead of "proper/appropriate").

We thank the Reviewer for this comment. The manuscript was double-checked by a native English-speaking colleague.
Due to the misinterpretation of the word proprietary (which meant created by the authors), we decided to change it to "original" to avoid confusing the reader.
We appreciate the Reviewer's knowledge of Polish syntax and thank for highlighting editorial errors.

What does "statistically significance difference" mean (e.g. line 198)? There are also numerous mistyping errors (e.g. lines 51-52 "detoriation" instead of "deterioration"; lines 55 doubling "our").

We thank the Reviewer for valuable comments. The mistyping and editorial errors were corrected.
We want to mention that “statistically significance difference” is often used in studies, even listed above by the Reviewer e.g., Symanzik C, Stasielowicz L, Brans R, Skudlik C, John SM. Prevention of occupational hand eczema in healthcare workers during the COVID-19 pandemic: A controlled intervention study. Contact Dermatitis. 2022 Dec;87(6):500-510. doi: 10.1111/cod.14206. Epub 2022 Aug 30. PMID: 35989622; PMCID: PMC9538141.
Nevertheless, we improved our manuscript due to the Reviewer’s comment, and changed indicated above forms.

Once again, as authors, we want to thank the Reviewer for the comments. We find them very valuable.

I am looking forward to review the improved version of this manuscript

Reviewer 2 Report

This article needs a slight change, typos, and spacing.

- 55 line "To the best of our our knowledge," word "our" was duplicated;

- in the introduction, talk more about what is hand eczema; atopic dermatitis; hand and nail psoriasis;

- the "ad" abbreviation, put in text, not only in the legend.

Author Response

RESPONSE TO REVIEWER 2:

The authors appreciate the thorough review and constructive comments. We have carefully assessed each of the comments and criticisms and we made changes to the manuscript accordingly. We believe that the revised manuscript has improved through this review process. We hope that the Editor and the Reviewers will agree with us. All changes to the manuscript after the Reviewer's suggestions are written in red.

We have provided a detailed response to Reviewer 2:

This article needs a slight change, typos, and spacing.

 - 55 line "To the best of our our knowledge," word "our" was duplicated;

We thank the Reviewer for this comment. We have removed the duplicated word.

- in the introduction, talk more about what is hand eczema; atopic dermatitis; hand and nail psoriasis;

We thank the Reviewer for this comment. We have added indicated information to the Introduction. We are pleased to be able to supplement our manuscript with such valuable data.

- the "ad" abbreviation, put in text, not only in the legend.

We thank the Reviewer for this comment. As we added the information about atopic dermatitis to the Introduction, the abbreviation AD is mentioned in this study section for the first time.

We truly appreciate the Reviewer’s comments and find them very valuable. 

Round 2

Reviewer 2 Report

No coments.